# Nutritional Niches of Cancer Therapy-Induced Senescent Cells

**DOI:** 10.3390/nu14173636

**Published:** 2022-09-02

**Authors:** Àngela Llop-Hernández, Sara Verdura, Elisabet Cuyàs, Javier A. Menendez

**Affiliations:** 1Girona Biomedical Research Institute, 17190 Girona, Spain; 2Metabolism and Cancer Group, Program Against Cancer Therapeutic Resistance (ProCURE), Catalan Institute of Oncology, 17005 Girona, Spain

**Keywords:** senescence, metabolism, glutamine, nutrition, cancer, miR146a

## Abstract

Therapy-induced senescence (TIS) is a state of stable proliferative arrest of both normal and neoplastic cells that is triggered by exposure to anticancer treatments. TIS cells acquire a senescence-associated secretory phenotype (SASP), which is pro-inflammatory and actively promotes tumor relapse and adverse side-effects in patients. Here, we hypothesized that TIS cells adapt their scavenging and catabolic ability to overcome the nutritional constraints in their microenvironmental niches. We used a panel of mechanistically-diverse TIS triggers (i.e., bleomycin, doxorubicin, alisertib, and palbociclib) and Biolog Phenotype MicroArrays to identify (among 190 different carbon and nitrogen sources) candidate metabolites that support the survival of TIS cells in limiting nutrient conditions. We provide evidence of distinguishable TIS-associated nutrient consumption profiles involving a core set of shared (e.g., glutamine) and unique (e.g., glucose-1-phosphate, inosine, and uridine) nutritional sources after diverse senescence-inducing interventions. We also observed a trend for an inverse correlation between the intensity of the pro-inflammatory SASP provoked by different TIS agents and diversity of compensatory nutritional niches utilizable by senescent cells. These findings support the detailed exploration of the nutritional niche as a new metabolic dimension to understand and target TIS in cancer.

## 1. Introduction

A well-established hallmark of cancer cells is their ability to evade senescence [1,2], a non-proliferative but metabolically active survival state that serves to prevent the propagation of genetically unstable, oncogenically-activated, and/or dysfunctional cells [3,4,5,6,7,8,9]. Tumor cells can, nevertheless, be forced into senescence by exposure to therapeutic agents, a phenomenon termed therapy-induced senescence (TIS). Through the expression and secretion of a variety of pro-inflammatory cytokines and chemokines, referred to as the senescence-associated secretory phenotype (SASP) [10,11,12,13,14], TIS cells represent a double-edged sword that promotes tumor elimination but also generates chronic inflammatory environments that can trigger or exacerbate cancer progression and normal tissue dysfunction [15,16,17,18,19].

TIS can render some cancer cells highly immunogenic while promoting the arrest of neighboring cancer cells, creating an environment that strongly activates anti-tumor immunity [20,21,22,23]. The long-term persistence of TIS in normal and tumor tissues may, however, be detrimental by increasing the likelihood of tumor relapse and undesirable adverse effects associated with cancer treatment. For instance, TIS cells that persist after therapy can elude immunosurveillance and accumulate at sites of aging pathologies [24,25,26]. They can also reprogram into stem-like states with associated drug resistance, which ultimately contributes to tumor relapse [27,28,29,30]. Similarly, the accumulation of TIS cells in inflammatory niches can activate epithelial-to-mesenchymal (EMT)-like programs that promote metastatic behavior in surrounding non-senescent cancer cells [30,31,32,33,34]. Finally, the local and systemic SASP signaling from persisting senescent cells in non-cancerous tissues actively contributes to short- and medium-term adverse reactions to cancer interventions in patients, including cardiac dysfunction, bone marrow suppression, bone loss, fatigue, and physical decline [15,35,36,37]. Indeed, studies in adult survivors of childhood cancer have highlighted several chronic gerontogenic effects of chemotherapy, including an increase in the prevalence of second malignancies, heart disease, and pulmonary events [38]. These observations led to the development of novel strategies aiming to selectively kill TIS cells (senolytics) and/or suppress all or at least some of their characteristics by blocking SASP (senostatics or senomorphics) [39,40,41,42,43,44,45].

A novel concept in TIS is the existence of distinct subtypes of senescent cells that can develop distinguishable phenotypes on the basis of epigenetic traits and environmental constraints [46,47,48,49]. Two major environmental variables are oxygen and nutrient availability, which vary among and within tissues. Oxygen is a known determinant for pro-inflammatory SASP expression, and physiological hypoxia can modulate the development of various senescence-associated phenotypes [50]. In relation to nutrient availability, we hypothesized that the scavenging and catabolic abilities of TIS cells would balance the fluctuations in available nutrient types and levels in their microenvironmental niches [51,52,53,54]. To provide a preliminary assessment of what nutrients are “*on the menu*” for TIS cells and to determine if they vary with mechanistically-diverse senescence-inducing interventions, we used Biolog Phenotype Microarrays (PM) to phenotype the preferred carbon and nitrogen sources utilized by TIS cells among 190 different nutritional sources under nutrient-poor conditions [55,56,57,58,59,60]. To assess senescence trigger-dependent effects, we investigated the utilization of different nutritional substrates by TIS cells and how they modulated the oxidation of major metabolic fuels (glucose and glutamine) to match nutrient availability while meeting energy demands. We then surveyed the metabolites that could be scavenged and catabolized by TIS cells as alternative fuels in the absence of glucose. Finally, we searched for correlations between the intensity of the pro-inflammatory SASP provoked by TIS agents and the diversity of compensatory nutritional niches utilizable by TIS cells.

## 2. Materials and Methods

### 2.1. Cell Lines and Culture

Human A549 lung cancer cells and embryonic kidney HEK293T cells were obtained from the ATCC (Manassas, VA, USA) and cultured in complete Dulbecco’s modified Eagle’s medium (Gibco/Invitrogen, Carlsbad, CA, USA), supplemented with 10% fetal bovine serum, 2 mmol/L L-glutamine, and 100 IU/mL penicillin-streptomycin (all from Gibco). All cells were tested for *Mycoplasma* contamination using a PCR-based assay prior to experimentation and were intermittently tested thereafter.

### 2.2. Senescence Induction

A549 cells were seeded in 150-mm dishes (800,000 cells/plate) for 24 h before treatment with bleomycin (20 μmol/L), doxorubicin (50 nmol/L), alisertib (500 nmol/L), or palbociclib (5 μmol/L). After 7 days of treatment, the cultures reached 70–80% confluency and were mostly senescent. Untreated (proliferative) controls (400,000 cells) were seeded in 150-mm dishes and cultured in parallel for 7 days, reaching ~90% confluency. For downstream applications, cells from one 150 mm dish were reseeded into 6-well (colony formation assays) or 96-well plates (metabolite phenotypic screening).

### 2.3. Colony Formation Assay

Proliferative and senescent cells (1000 cells/well) were grown for 10 days in a drug-free medium. Colonies were fixed with 4% paraformaldehyde (*v/v*) and stained with 0.5% crystal violet (*w/v*) for visualization.

### 2.4. Staining for Senescence-Associated β-Galactosidase Activity

Cell β-galactosidase activity was detected with the Senescence β-Galactosidase Staining Kit (Cat. #9860, Cell Signaling Technology, Danvers, MA, USA).

### 2.5. Biolog Metabolite Phenotypic Screening

In total, 50 μL/well of 400,000 cells/mL suspensions of TIS and proliferative cells (20,000 cells per well) in Biolog IF-M1 medium (RPMI-1640 medium lacking phenol red and depleted of carbon-energy sources, low glutamine [0.3 mmol/L], and low FBS [5%]), were transferred to Phenotype PM-M1 and PM-M2 MicroArrays (Biolog, Hayward, CA, USA) containing 190 biochemical substrates that could potentially be metabolized and provide energy for cells. The content of plates PM-M1 and PM-M2 can be found at https://www.biolog.com/wp-content/uploads/2020/04/00P-134-Rev-F-PMM-MicroArrays-Brochure-PM-M1-to-PM-M14.pdf (accessed on 28 August 2022). After a 48 h incubation, a period that allows cells to use any residual carbon energy sources in the 5% serum (~0.35 mmol/L glucose plus lipids and amino acids) and minimizes the background color in the negative control wells that have no added biochemical substrate, the respective utilization of substrates to generate NADH was measured as an optical density (OD) at 590 nm for 1.5 h and imaged.

### 2.6. Mitochondrial Fuel Oxidation Analyses

Oxidation rates of glucose and glutamine were measured using an XFp Extracellular Flux Seahorse Analyzer (Agilent Technologies, Palo Alto, CA, USA). The Agilent Seahorse XF Mito Fuel Flex Test Kit was used in a standard protocol https://www.agilent.com/cs/library/usermanuals/public/XF_Mito_Fuel_Flex_Test_Kit_User_Guide%20old.pdf (accessed on 4 August 2022). The test measures the dependency, capacity, and flexibility of cells to oxidize three mitochondrial fuels in real-time: glucose (pyruvate), glutamine (glutamate), and long-chain fatty acids. The test determines the rate of oxidation of each of these fuels by measuring the oxygen consumption rate in the presence or absence of fuel pathway inhibitors: the glucose oxidation pathway inhibitor UK5099 (2 μmol/L), which blocks the mitochondrial pyruvate carrier; the glutamine oxidation pathway inhibitor BPTES (3 μmol/L), which allosterically inhibits glutaminase-1; and the long-chain fatty acid oxidation inhibitor etomoxir (4 μmol/L), which inhibits carnitine palmitoyl-transferase 1A, a critical enzyme of mitochondrial β-oxidation. Seahorse Wave Desktop software was used for data generation and analysis, and GraphPad PRISM 8 (San Diego, CA, USA) was used for statistical analysis and data presentation.

### 2.7. Lentiviral Transduction

The PHAGE-PmiR-146a-GFP-PGK-puro plasmid was a kind gift from Stephen Elledge (Department of Genetics, Harvard Medical School, Division of Genetics, Brigham and Women’s Hospital, Howard Hughes Medical Institute, Boston, MA, USA). Viral particles were produced in 293T cells by co-transfection of the PHAGE-PmiR-146a-GFP-PGK-puro plasmid with a 3rd generation lentivirus packaging system, consisting of pCMV-VSV-G (Addgene, Cambridge, MA, USA, #8454) and pCMV-dR8.2 dvpr (Addgene, Cambridge, MA, USA, #8455). Transient transfection was performed in 293T cells to produce lentiviral supernatants, and A549 cells were infected with lentiviral supernatants using 8 μg/mL polybrene. After a 48 h incubation, the supernatant was replaced by a medium containing 10 μg/mL puromycin for a further 48 h.

### 2.8. IncuCyte Assay

Cells (2000/well) were seeded into 96-well plates and cultured up to 10 days. Phase and green fluorescence (400 ms acquisition) images were collected at 37 °C every 4 h over 10 days using the IncuCyte Zoom (IncuCyte S3; Essen BioScience, Ann Arbor, MI, USA).

### 2.9. Flow Cytometry

miR146a-EGFP reporting-containing A549 cells were harvested and suspended in 300 μL of a medium. GFP positivity was determined using a 488 nm laser excitation wavelength (FITC channel) on a BD Accuri C6 Flow Cytometer (BD Biosciences, San Jose, CA, USA). Data were analyzed using FCS Express 7 software (De Novo™ Software, Pasadena, CA, USA) and were depicted as mean fluorescence intensities (arbitrary units) of three independent experiments.

### 2.10. Data Handling

To circumvent the lack of well-established analytical and statistical methodology to analyze the endpoint mode of Biolog PM plates, including systematic biases towards increased or decreased metabolic signals due to various causes (e.g., plate batch inter-variability and color development in the negative-control wells due to substrate reservoirs) [59], experimental data were not pooled from independent readings. Rather, the data shown are from a representative experiment (from at least two separate experiments for each TIS trigger and performed with *n* = 3 replicate measurements) performed in a single run. Arbitrary thresholds were set to disregard changes that commonly arise when performing pairwise two-tailed *t*-tests on color intensity readings, especially when the arrays are dominated by non-active profiles. All nutritional sources that exceeded the established thresholds (±1.5 or larger [±2.0]) for each pair of TIS/proliferative cells were included in the flower model Venn diagrams when qualitative substrate discrepancies did not exist between independent replicates. Data from Seahorse-based mitochondrial fuel oxidation analyses and IncuCyte assays are presented as mean ± S.D.

## 3. Results

### 3.1. Establishment of Therapy-Induced Senescence Models

We first established the conditions to screen carbon and nitrogen sources for TIS cells using A549 cells, a widely used model of lung adenocarcinoma and an *in vitro* model of type II pulmonary epithelial cells [61,62]. Cells were treated with increasing concentrations of four clinically-relevant senescence triggers: (1) Bleomycin, an anti-tumor antibiotic that promotes oxidative DNA damage mediated by reactive oxygen species (ROS), and is commonly used to treat Hodgkin’s lymphoma and testicular germ-cell tumors and to induce pulmonary fibrosis in cancer-free mice [61]; (2) Alisertib (MLN8237), a selective aurora A kinase inhibitor that is under investigation for several malignancies, including hematologic (non-Hodgkin’s lymphoma) and solid tumors [42,63]; (3) Doxorubicin, a topoisomerase II inhibitor used to treat a variety of cancers including lung, breast, lymphoma, and acute lymphocytic leukemia [64,65,66]; (4) Palbociclib, a selective inhibitor of cyclin-dependent kinases 4 and 6 (CDK4/6) that is approved for the treatment of advanced or metastasized estrogen receptor (ER)-positive and epidermal growth factor receptor 2 (HER2)-negative breast cancer [67,68,69]. These treatments (for up to 7 days) resulted in decreased confluency and cell proliferation arrest, and the so-called optimal senescence-inducing concentrations were selected as those allowing cultures to reach about 70–80% confluency while promoting the occurrence of the major classical markers of senescence, namely an enlarged and flattened cell shape and enhanced senescence-associated β-galactosidase (SA-β-gal) activity [9,70,71], in the highest number of cells. Treatment with optimal senescence-inducing concentrations of bleomycin (20 μmol/L), doxorubicin (50 nmol/L), and alisertib (500 nmol/L) resulted in 80–100% SA-β-gal-positive populations (Figure 1A, top). Treatment with palbociclib (5 μmol/L) resulted in 60–70% SA-β-gal-positive populations (Figure 1A, top).

To test the stability of the senescent state, untreated and TIS cells (7-day-treatment) were seeded at low-density (1000 cells/well) under drug-free conditions during 10 days to allow the formation of colonies. TIS cells previously exposed to bleomycin, alisertib and doxorubicin showed no clonogenic capacity, demonstrating the stability of the senescent phenotype (Figure 1A, bottom). By contrast, palbociclib-induced senescent cells showed a partial recovery of proliferation potential upon drug release (Figure 1A, bottom).

### 3.2. Senescence Trigger-Dependent Metabolic Fingerprints

Proliferative and TIS cells grown in parallel were simultaneously plated onto PM-M1 and PM-M2 microplates with wells coated with substrate nutrients that could serve as carbon and/or nitrogen sources, creating 190 unique nutritional “niches” comprising carbohydrates, nucleic acids, glycosylamines, metabolic intermediates, amino acids, and dipeptides [55,56,57,58,59,60] (Figure 1B). Blank wells and wells pre-coated with glucose were included as negative and positive controls, respectively. The proliferative and TIS cells were incubated under nutrient-limiting conditions with a Biolog IF-M1 medium (RPMI 1640, no glucose, 0.3 mmol/L glutamine, and 5% serum), providing all nutritional requirements at sufficient levels other than major carbon and nitrogen sources, which were omitted. Assays were conducted over 48 h.

To assess senescence trigger-dependent effects, we first determined the substrate utilization of paired proliferative/TIS cells in each nutritional niche using a colorimetric dye that is reduced when cells catabolize the extracellular substrate and generate energy-rich NADH. The optical density (OD) values of each substrate at 590 nm (purple color) resulting from the accumulation of reduced dye over a 90 min period were normalized to those of the negative-control wells included in each microplate (Figure 2A; source data are available online at Appendix A). The PM-M1 plate contained primarily carbohydrate and carboxylate substrates, whereas the PM-M2 plate contained individual L-amino acids and dipeptide combinations. We then calculated the absolute ratio between normalized OD values of senescent versus proliferative states. Detailed consumption maps of nutritional substrates that showed ±1.5 or larger (±2.0) fold-changes for each pair of TIS/proliferative cells are shown in Appendix A. Flower model Venn diagrams were generated to identify the core and differential carbon/nitrogen sources under- or over-utilized (±2.0 fold-changes) by the four TIS-generated phenotypes (Figure 2B). All TIS cells shared a reduced ability to metabolize mono- (glucose and mannose), di- (maltose), and tri- (maltotriose) saccharides compared with matched proliferative controls. Most TIS-generated types also exhibited a reduced capacity to metabolize the glucose polysaccharide, dextrin. Notably, TIS cells generated in response to palbociclib, doxorubicin, and alisertib showed an enhanced capacity (>2.0-fold) to utilize glutamine-containing dipeptides compared with matched proliferative controls (Figure 2B). All TIS-generated types shared an ability to overutilize glutamine-containing peptides compared with proliferative counterparts when using a fold-change cut-off of >1.5. 

Our findings thus far suggest that TIS cells preferentially use glutamine over glucose as mitochondrial fuel under nutrient-limiting conditions. To test this hypothesis, we measured the dependency, capacity, and flexibility of proliferative and TIS cells (doxorubicin- and alisertib-induced) to oxidize glucose and glutamine using the Seahorse XF Mito Fuel Flex Test (Figure 2C). Compared with proliferative cells, TIS cells had a notably lower capacity to use and increase the oxidation of glucose when trying to compensate for the inhibition of glutaminolysis and long-chain fatty acid oxidation as alternative fuel pathways using glutaminase-1 inhibitor BPTES and carnitine palmitoyltransferase-1 inhibitor etomoxir, respectively (Figure 2C). Conversely, no remarkable differences were observed in the ability of proliferative and TIS cells to employ and augment the oxidation of glutamine when trying to compensate for inhibition of glucose and long-chain fatty acid oxidation using the mitochondrial pyruvate carrier inhibitor UK5099 and etomoxir, respectively (Figure 2C).

### 3.3. Alternative Metabolic Fuels of Therapy-Induced Senescent Cells in the Absence of Glucose

We normalized all the data to the positive control (wells pre-coated with glucose) to better understand the flexibility of TIS cells in utilizing alternative carbon/nitrogen fuel sources in response to the severe hypoglycemic/aglycemic conditions of the PM assays (Appendix A). Flower model Venn diagrams were used to identify core and differential carbon/nitrogen sources capable of rescuing the maximum glucose catabolism. Using a 20% cut-off (i.e., rescue of at least one-fifth of the glucose capacity; Figure 3A), we found that all TIS phenotypes utilized the following core nine metabolites in the absence of glucose: the six-carbon sugar fructose-6-phosphate, the keto-acid pyruvic acid, the amino acid glutamine, and numerous dipeptides containing this amino acid (Ala-Gln, Arg-Gln, Asp-Gln, Gln-Glu, Gln-Gln, and Gln-Gly). The glycolytic intermediate glucose-6-phosphate also substituted for glucose for a majority of TIS-generated phenotypes. The nucleoside inosine also supported the bioenergetic activity of bleomycin- and doxorubicin-induced senescent cells in the absence of glucose. Doxorubicin-induced senescent cells showed a differential capacity for bioenergetic activity in the absence of glucose, with the nucleoside uridine, the lipid biosynthesis precursor α-glycerol-phosphate, the succinic acid derivatives succinamic acid and mono-methyl succinate, and the alanine-containing dipeptides Ala-Pro, Ala-Tyr, as alternative metabolic substrates. After applying a more stringent 25% cut-off (i.e., rescue of at least one-quarter of the glucose capacity; Figure 3B), most of the TIS-generated phenotypes (doxorubicin-, alisertib-, and palbociclib-induced) preferentially catabolized glutamine-containing dipeptides and pyruvic acid for growth in the absence of glucose. Glucose-1-phosphate remained as one of the metabolites capable of rescuing glucose in doxorubicin- and palbociclib-induced senescent cells, irrespective of the cut-off level.

None of the candidate metabolites that supported the viability of TIS cells under glucose-deficient/nutrient-limiting conditions were related to the stimulation of proliferative subpopulations following exposure to different senescence inducers. When proliferative counterparts were included in the flower model Venn diagrams, all the core nutrients identified as rescuing cells in the absence of glucose (i.e., dextrin, glycogen, maltotriose, maltose, mannose, and glucose itself) were distinct from those differentially utilized by TIS cells, irrespective of the applied cut-off level (Figure 3A,B).

### 3.4. Correlation between SASP Activity and Nutritional Niche Diversity in Therapy-Induced Senescent Cells

Finally, we sought to establish whether the activation level of SASP correlated with the diversity of nutritional niches utilized by TIS cells. The microRNA, miR146a, is expressed in response to elevated inflammatory cytokine levels as part of a negative feedback loop that restrains excessive SASP activity [63,72,73]. We thus engineered A549 cells expressing a senescence-reporter construct containing the miR146a promoter linked to EGFP (Figure 4A) as a surrogate marker of miR146a induction following treatment of cells with the panel of senescence inducers, and utilized the IncuCyte^®^ Live-Cell Analysis System to monitor real-time reporter activity (up to 10 days) (Figure 4A). Notably, we found significant differences in the ability of senescence-inducing agents to induce the reporter in a concentration- and time-dependent manner. Bleomycin was the most powerful inducer, followed by alisertib, palbociclib, and doxorubicin, with the latter showing the least potency in A549 cells (Figure 4B, left panel). Flow cytometry analyses confirmed that bleomycin-treated cells showed a higher miR146a-related mean fluorescence intensity (MFI) than doxorubicin-treated ones (data not shown).

Doxorubicin-induced senescent cells could utilize the greatest number of metabolites (21), followed by those induced by alisertib (14), palbociclib (13), and bleomycin (12), using a 20% cut-off value. With a more stringent 25% cut-off, doxorubicin-induced senescent cells remained the most flexible in terms of utilizable metabolites (10), followed by palbociclib (9), alisertib (6), and bleomycin (3) treatments. Thus, whereas a fewer number of metabolites could be used to overcome glucose/nutrient deprivation in bleomycin-induced senescent cells, which exhibited the greatest SASP activity (as measured by the reporter construct), notably higher substrate diversity was evident in palbociclib- and doxorubicin-induced senescent cells, which showed lower SASP activity (Figure 4B, right panel).

## 4. Discussion

Understanding the metabolic needs of cells that undergo TIS might help determine what drives their paradoxical effects in vivo and provide new opportunities for therapy [74,75,76,77]. We hypothesized that TIS cells under prolonged nutrient fluctuations metabolically adapt their scavenging and/or catabolic ability. We tested this in nutrient-limiting conditions using an array format with almost 200 nutrients [55,56,57,58,59,60] to delineate the metabolic fingerprints associated with mechanistically-diverse senescence triggers. We provide evidence of distinguishable TIS-associated nutrient consumption profiles involving a core set of shared, but also of unique, nutritional sources after different senescence-inducing interventions. We suggest that the flexibility of TIS cells to use compensatory metabolites under glucose-deficient conditions inversely correlates with the intensity of the SASP provoked by TIS agents.

In a comparative analysis with proliferating cells, we evaluated the types of nutritional substrates differentially utilized by TIS cells established under the four treatment regimens, which revealed a shared bioenergetic under-utilization of glucose and glucose-based saccharides and an over-utilization of glutamine. This suggests a common metabolic reorganization in TIS cells of reduced oxidation of glucose and increased dependency on glutamine. By measuring mitochondrial fuel usage, we confirmed that TIS cells displayed a notably lower requirement and ability to use glucose to meet metabolic demand when attempting to compensate for BPTES-induced inhibition of glutamine oxidation. Our findings are in line with previous studies that showed that glutamine consumption and metabolism increase in several senescence states. For instance, senescence of human fibroblasts induced by Nutlin3a, oxidative, replicative, and oncogene-related causes is obligatorily accompanied by the up-regulation of glutaminase 1, a key mitochondrial enzyme in the glutaminolysis pathway that converts glutamine into Krebs cycle metabolites [78]. Mitochondrial glutamine anaplerosis has also been shown to mediate senescence induction following chemotherapy-induced DNA damage [79]. Intriguingly, the small subset of cancer stem-like cells that can evade TIS after long periods of persistence in a dormant state appear to similarly rely on glutamine metabolism [80,81]. These findings, altogether, strongly support the targeting of glutaminolysis as a promising senolytic strategy for both TIS cells and dormant senescent cells fueling tumor relapse [30].

We investigated both the common and exclusive nutrients that could serve as alternative sources under glucose/nutrient-restricted conditions to phenotype the metabolic processes of TIS cells. Glutamine and glutamine-containing peptides formed the overlapping core of alternative nutrients scavenged and catabolized by all types of TIS cells studied. From an ecological perspective, the metabolic redundancy within TIS “communities” indicates that individual “species” of TIS cell may not occupy a specific metabolic niche (Figure 5). It should, however, be acknowledged that some types of TIS cells utilized carbon/nitrogen sources that were not used by others, which hints at a senescence trigger-dependent occupation of specific nutritional niches.

Glucose 1-phosphate, a key intermediate of the glucuronic acid pathway involved in the conversion of galactose to glucose, synthesis of glycogen and glycosaminoglycans, and metabolism and clearance of chemotherapeutic drugs via glucuronosylation [82,83,84,85,86], was differentially employed by doxorubicin- and palbociclib-induced senescent cells. The non-adenosine nucleosides, inosine and uridine, were exclusive nutritional sources for senescent cells induced by the DNA-damaging drugs bleomycin and doxorubicin. Extracellular inosine has recently been shown to relieve tumor-imposed metabolic restrictions on T-cells by operating as an alternative substrate for cell growth and function of effector T-cells in the absence of glucose [87]. Extracellular uridine has been shown to rescue nutrient stress (glucose deprivation) in astrocytes and neurons [88,89], and was more recently reported to be a compensatory metabolite in pancreatic cancer cells growing under nutrient-deficient conditions [60]. The ribose moiety of inosine and uridine can be used as a carbon source for central metabolic pathways, providing not only biosynthetic precursors for anabolic proliferation (e.g., via nucleotide salvage), but also fueling energetic metabolism via routing through the pentose-phosphate pathway and subsequent oxidative catabolism [60,87]. The upregulation of nucleoside usage in diverse cell types including immune, cancer, and now TIS cells, indicates that metabolic competition impacting both immunosuppression and tumor progression might occur in interstitial tumor fluid where uridine levels are highly enriched (up to millimolar levels) relative to their levels in serum [51]. Notably, suppression of nucleotide synthesis, particularly that of pyrimidines such as uridine, is a critical metabolomic adaption that plays a causative role in the establishment of replicative- and oncogene-induced senescence [90,91]. Moreover, exogenous uridine has recently been described as a pro-regenerative metabolite capable of reducing physiological and pathological stem cell senescence and promoting tissue regeneration and repair [92]. As TIS is not necessarily detrimental, the ability of DNA damage-induced senescent cells to utilize nucleosides for cellular energetic needs in the absence of sufficient glucose might be understood in terms of facilitating repair (including DNA repair) in tissues injured by cancer therapy [93,94,95]. Although we recognize the limitations of extrapolating our *in vitro* findings with a lung adenocarcinoma cell line to TIS in normal tissues, the senescence response of A549 cells to DNA damage agents such as bleomycin was not very different from that observed in normal alveolar epithelial cells [61].

Delayed induction of miR146a is a compensatory response that restrains the pro-inflammatory expression and secretion of IL-6/IL-8 driven by the robust activation of the IL-1α signaling pathway [72,73]. Thus, whereas ectopic expression of miR146a down-regulates critical components of the IL-1α receptor pathway and suppresses IL-6/IL-8 secretion, cells undergoing senescence without induction of a robust SASP do not express miR146a. Because this safeguard may be particularly important when the local concentration of SASP factors is high—for example, when senescence is induced after exposure to chemotherapeutic agents—it is noteworthy that the higher the intensity of SASP-driven induction of miR146a, the lower the number of compensatory nutritional niches utilizable by TIS cells. This inverse correlation suggests that the SASP-related potency of a given TIS agent might fine-tune the metabolic flexibility of the generated senescent cells to acquire and use nutrients efficiently, thereby dictating the number and successful occupancy of discrete nutritional niches (Figure 5). Mechanistically, if greater metabolic inflexibility relates to disrupted inflammatory assuagement, we speculate that a certain degree of metabolic flexibility, as an adaptation to energy resources and requirements of TIS cells, is necessary for the mitigation of the SASP-related inflammatory processes. A majority of previous studies have focused their efforts on the deregulation of nutritional sensors (e.g., mTOR, AMPK, etc.) in the establishment of the senescent phenotype [96,97,98,99,100] and how correcting those nutrient sensing pathways could promote senescent cell death [101,102,103,104]. However, much less is known about the metabolic features, plasticity, and adaptation of TIS cells to their respective microenvironments [30]. We now propose that unveiling the key drivers linking SASP with the degree of metabolic (in)flexibility of TIS might reveal new targets for intervention strategies aimed to clinically manage tumor relapse and adverse side-effects in cancer patients.

## 5. Conclusions

This is the first report, to our knowledge, showing that utilization patterns of different carbon/nitrogen energy sources under compromised nutritional conditions vary across different types of TIS cells, which would likely make them differentially responsive to changes in their nutritional microenvironment. Whether the specific metabolic fingerprints of TIS can be linked to tumor relapse and/or adverse effects warrants further investigation, for example, using senescent cells from patient tissues. Nonetheless, application of the Biolog metabolite phenotypic screening platform [96] enabled the rapid acquisition of source-agnostic information about the nutrients utilized by TIS cells, which may enable a more detailed exploration of the nutritional niches to understand and target TIS in cancer.

## Figures and Tables

**Figure 1 nutrients-14-03636-f001:**
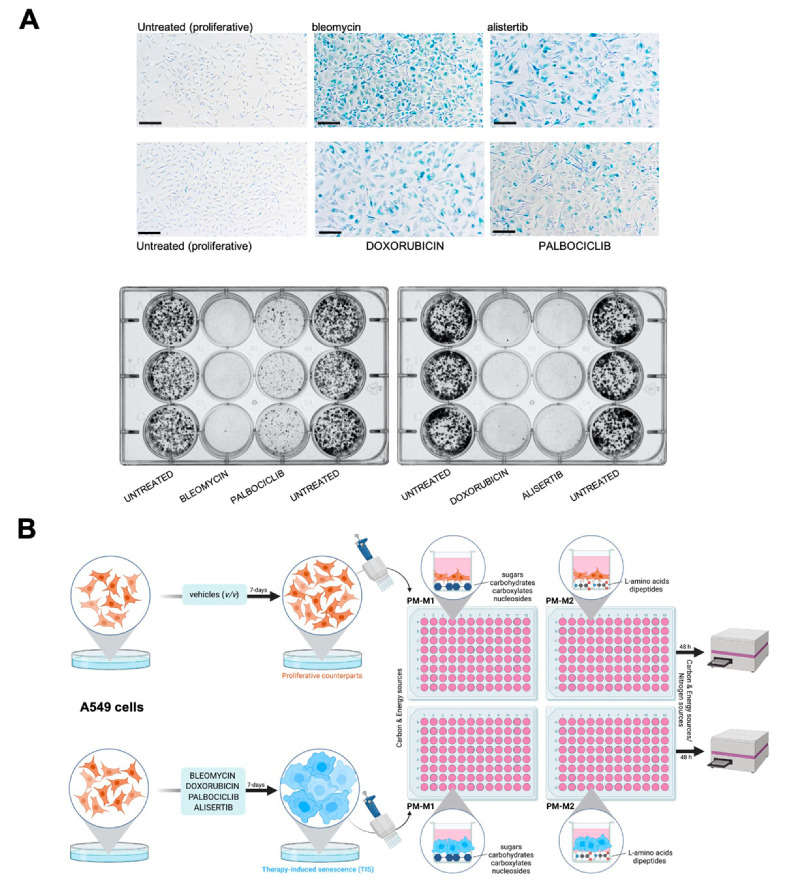
**Metabolic fingerprinting of therapy-induced senescence: an experimental approach.** (**A**) A549 cells were cultured for 7 days with bleomycin (20 μmol/L), alisertib (0.5 μmol/L), doxorubicin (50 nmol/L), and palbociclib (5 μmol/L). Top: representative images of SA-β-gal staining from three independent experiments. Scale bar: 200 μm. Bottom: representative images from 6-well plates of 10-day clonogenic survival analyses of A549 cells previously cultured for 7 days with therapy-induced senescence agents. (**B**) Schematic representation of metabolite utilization analysis workflow in proliferative versus TIS cells using the Phenotype MicroArrays PM-M1 and PM-M2.

**Figure 2 nutrients-14-03636-f002:**
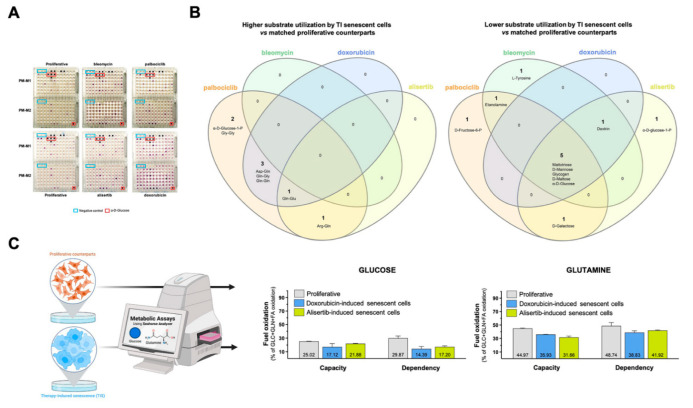
**Substrate utilization patterns of therapy-induced senescence cells.** (**A**) Representative images of paired proliferative/TIS cells assayed in PM-M1 and PM-M2 plates. Negative control wells (blue boxes) have no substrate. Wells containing D-glucose (red boxes) served as positive controls. (**B**) Flower model Venn diagrams showing higher (left) or lower (right) substrate utilization in each type of TIS cell. (**C**) Analysis of mitochondrial oxidation of glucose and glutamine in proliferative and TIS cells using the Agilent Seahorse XF Mito Fuel Flex kit.

**Figure 3 nutrients-14-03636-f003:**
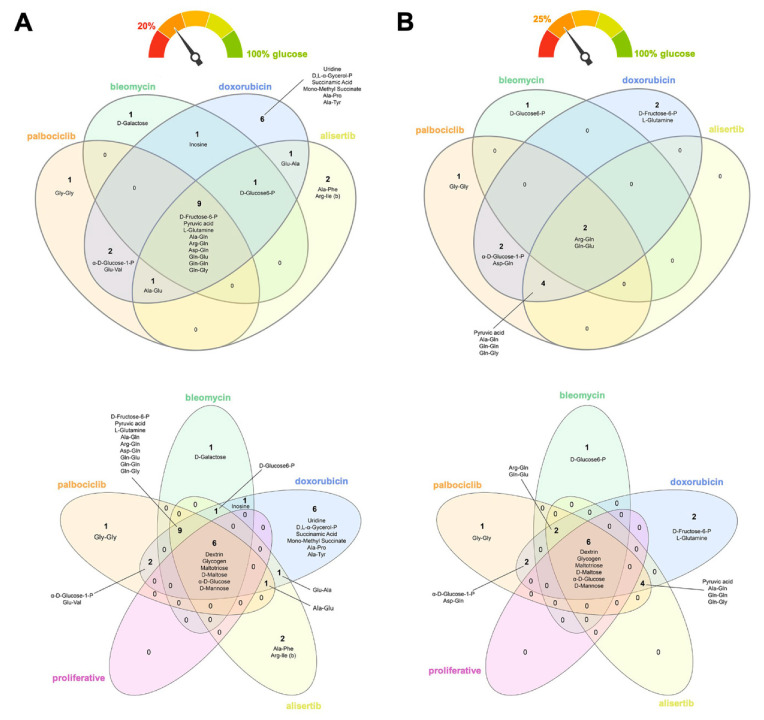
**Nutritional rescue of glucose deprivation in therapy-induced senescence cells.** Flower model Venn diagrams showing shared and unique metabolic substrates capable of rescuing at least (**A**) one-fifth or (**B**) one-quarter of the bioenergetic capacity of glucose in TIS cells.

**Figure 4 nutrients-14-03636-f004:**
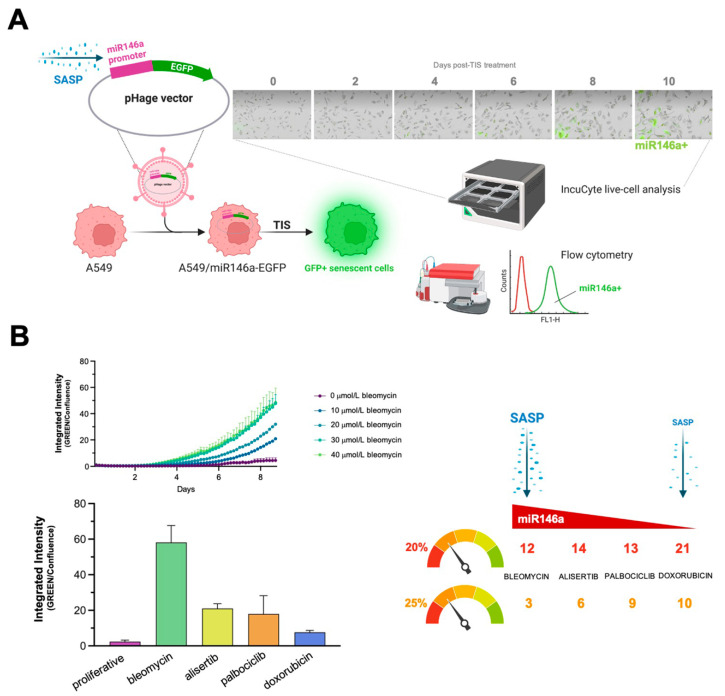
**Activation levels of the negative-feedback regulator of the SASP response, miR-146a, and diversity of nutritional rescue of glucose deprivation in therapy-induced senescence cells.** (**A**) The miR146a-EGFP reporter detects senescence in A549 cells. The SASP-responsive miR146a-EGFR reporter was transfected into A549 cells and TIS was induced through treatment with bleomycin, alisertib, doxorubicin, and palbociclib. EGFP fluorescence was measured using either IncuCyte Zoom or flow cytometry. Acquisition and analysis of images was carried out using fully integrated algorithms in an IncuCyte S3 analysis system. (**B**) Left: kinetic plots of SASP-driven miR146a-EGFP reporter expression following senescence induction with bleomycin, alisertib, doxorubicin, and palbociclib. Right: correlation between SASP intensity and number of metabolites (as calculated in Figure 3A,B) circumventing glucose-deprived conditions following senescence induction with bleomycin, alisertib, doxorubicin, and palbociclib.

**Figure 5 nutrients-14-03636-f005:**
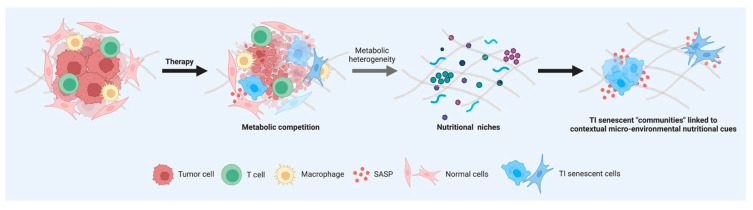
**Nutritional fingerprints of therapy-induced senescent cells: linking senescent phenotypes to metabolic niches.** Cells in a TIS state share several common features, such as changes in morphology, increase in β-galactosidase activity, and activation of an inflammatory response. Understanding whether all types of senescent cells induced by exogenous therapeutic stresses are identical or heterogeneous in terms of their metabolic needs might aid in achieving the goal of selectively eliminating the deleterious effects of TIS cells. The recognition of inter-TIS heterogeneity in terms of scavenging and catabolic ability contingent on nutrient availability might help to molecularly understand and therapeutically exploit TIS phenotype-associated “communities” linked to contextual micro-environmental nutritional cues in complex cancer and normal tissues.

## Data Availability

All data generated or analyzed during this study are included in this published article (and its Appendix A).

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
