# Peer review of "Nutritional Niches of Cancer Therapy-Induced Senescent Cells"

_nutrients, 2022, doi:10.3390/nu14173636_

Round 1
Reviewer 1 Report
Therapy-induced senescence (TIS) is a state of stable proliferative arrest of both normal and neoplastic cells that is triggered by exposure to anticancer treatments. TIS cells acquire a senes-cence-associated secretory phenotype (SASP), which is proinflammatory and actively promotes tumor relapse and adverse sideeffects in patients.
--Comment: Appropriately explained in Intro.
The authors hypothesized that TIS cells adapt their scavenging and/or catabolic ability to overcome the nutritional constraints in their microenvironmental niches. They used a panel of mechanistically-diverse TIS triggers and Biolog Phenotype MicroArrays to identify (among 190 different carbon and nitrogen sources) candidate metabolites that support the survival of TIS cells in limiting nutrient conditions.
--Comment: Here the raw data are missing and a full list of the 190 carbon and nitrogen sources. If you think you have included this already, please stress this a bit more in the results section. Furthermore, provide a supplement with the full original data. Do also statistics, what are significant measurements etc.
We provide evidence of distinguishable TIS-associated nutrient consumption profiles involving a core set of shared (e.g., glutamine) and more unique (e.g., glucose-1-phosphate, inosine and uridine) nutritional sources after diverse senescence-inducing interventions. We also observed a trend for an inverse correlation between the intensity of the pro-inflammatory SASP provoked by different TIS agents and the diversity of compensatory nutritional niches utilizable by senescent cells. These findings support the detailed exploration of the nutritional niche as a new metabolic dimension to understand and target TIS in cancer.
--Comment: Is reasonable well done, however, discussion should stress a bit more earlier findings and publications supporting these observations.
Reviewer 2 Report
The authors hypothesized that TIS cells adapt their scavenging and/or catabolic ability to overcome the nutritional constraints in their microenvironmental niches. They used a panel of mechanistically-diverse TIS triggers and Biolog Phenotype MicroArrays to identify (among 190 different carbon and nitrogen sources) candidate metabolites that support the survival of TIS cells in limiting nutrient conditions.
Moreover, the authors showed for the first time that utilization patterns of different carbon/nitrogen energy sources under compromised nutritional conditions vary across different types of TIS cells, which would likely make them differentially responsive to changes in their nutritional microenvironment. The research work is very innovative and interesting with a great impact on practical applications. So, the manuscript is presented very well and I propose only minor revisions:
1) To a better evaluation of results due to the proliferation after some treatments and/or co-transfection, it can be useful to report also the evaluation of a proliferation assay (as MTT or equivalent assay) and/or cytotoxicity assay (as lactate dehydrogenase) as control.
2) It need to insert the numbers of Figure captures under each figure
3) It need to substitute in text (Fig. 5) with (Figure 5) for the whole uniformity of the paper, and to correct β-galactosidase in figure capture of Fig 5, instead of @-galactosidase.
